# Transboundary Fisheries Management in Kavango–Zambezi Transfrontier Conservation Area (KAZA-TFCA): Prospects and Dilemmas

**Imikendu Imbwae [1,2,*], Shankar Aswani [1,3], Warwick Sauer [1] and Clinton Hay [4,5]**

1    Department of Ichthyology and Fisheries Science (DIFS), Rhodes University, Makhanda 6140, South Africa
2    Department of Fisheries, Choma P.O. Box 630450, Zambia
3    Department of Anthropology, Rhodes University, Makhanda 6140, South Africa
4    Namibia Nature Foundation, Windhoek 10005, Namibia
5    Environmental Sciences, University of Namibia, Windhoek 13301, Namibia
*    Correspondence: sundayimikendu@yahoo.com

**Abstract:** Inland fisheries in the Kavango–Zambezi Transfrontier Conservation Area (KAZA-TFCA) offer food security to the riverine communities across the region. They also contribute towards the attainment of the United Nations' Sustainable Development Goals 1 and 15, which aim to alleviate poverty and maintain biodiversity conservation. Despite this significant role, the fisheries have suffered severe declines in the previous decades due to multiple factors, such as overfishing and poor legislation. Furthermore, climate change is exerting pressure by altering the ecology and productivity of the river systems. The unprecedented challenges of the COVID-19 pandemic have further constrained management efforts. Attempts to address these challenges have pointed towards transboundary fisheries management as a silver bullet in moving towards sustainable fisheries management. However, the implementation of this strategy in the region has encountered numerous roadblocks, thereby subjecting the river ecosystem to a wider environmental threat, with dire consequences on livelihoods. This paper reviews existing management and governance structures together with key informant interviews to elicit primary and secondary data essential for management at the regional level. The study identifies conflicting regulations, and inadequate policies and institutions across the region as major bottlenecks affecting the successful implementation of transboundary fisheries management. Finally, the paper offers some suggestions for the improvement of fisheries management in the region.

**Keywords:** governance; inland fisheries; transboundary; Kavango–Zambezi; CCRF





## 1. Introduction

Inland fisheries in the KAZA-TFCA play a multifaceted role in providing food security, employment opportunities, and a cheap source of protein [1–3]. However, over the last half-century, there have been considerable reports of fish decline and shifts in fish species composition from inland water sources [4,5]. Studies suggest that the biodiversity crisis is more severe today in inland ecosystems than terrestrial ones, partly due to overfishing, weak institutions, and habitat degradation [5]. These combined and synergistic effects have negatively affected the abundance and range of inland fisheries [6–8]. Furthermore, given the acknowledged acceleration of climate change, weather patterns are expected to be more variable and unpredictable, thereby creating stochastic changes that will alter the ecology and ecosystem processes of river systems [9–11]. KAZA-TFCA exemplifies this sustainable challenge [12]. The area is drained by the palaeo-evolutionary linked Kavango and Zambezi River systems, which cover five countries, namely Angola, Botswana, Namibia, Zambia, and Zimbabwe (Figure 1). Hackenberg et al. [12] notes that the changes in flood patterns over the Zambezi River basin imply that the natural characteristics of these

aquatic resources are constantly changing with regards to functional responses of fishes, such as distribution and migration patterns. In addition to these problems, the COVID-19 pandemic has constrained management efforts of various countries to engage in timely fisheries management programs [13]. Aquaculture has been widely identified as a possible intervention to resuscitate inland fisheries in the region [14]. However, the prospects of the aquaculture sector are threatened by several factors, such as deterioration of pure brood stock due to the introduction of invasive fish species, such as *Oreochromis niloticus*, which has the capability to distort native *Oreochromis* fish species through hybridization and predation. Other invasive fish species, such as crayfish (*Cherax quadricarinatus*), may be risk factors (vectors) for the transfer of fish diseases and parasites [15]. Several studies have revealed that most of the fisheries in the KAZA-TFCA have experienced severe declines in fish catch rates [16–18]. The KAZA-TFCA is a regional and international collaborative initiative to effectively manage natural resources, such as wildlife, fisheries and forestry, that straddle across boundaries in line with the Southern Africa Development Community (SADC) protocol of 2001 [19]. Fish decline in this region is a serious concern among different actors along the agri-fisheries value chain [20,21]. For nearly over a decade now, transboundary fisheries management efforts have attempted to scale up conservation to a larger scale [22–24]. However, these efforts have encountered numerous roadblocks, and fish decline and illegal fishing practices are still widely reported [8,17,25,26]. As the population increases and exerts pressure on the entire Kavango–Zambezi River system, the need to have a viable transboundary fisheries management regime is crucial [27,28].

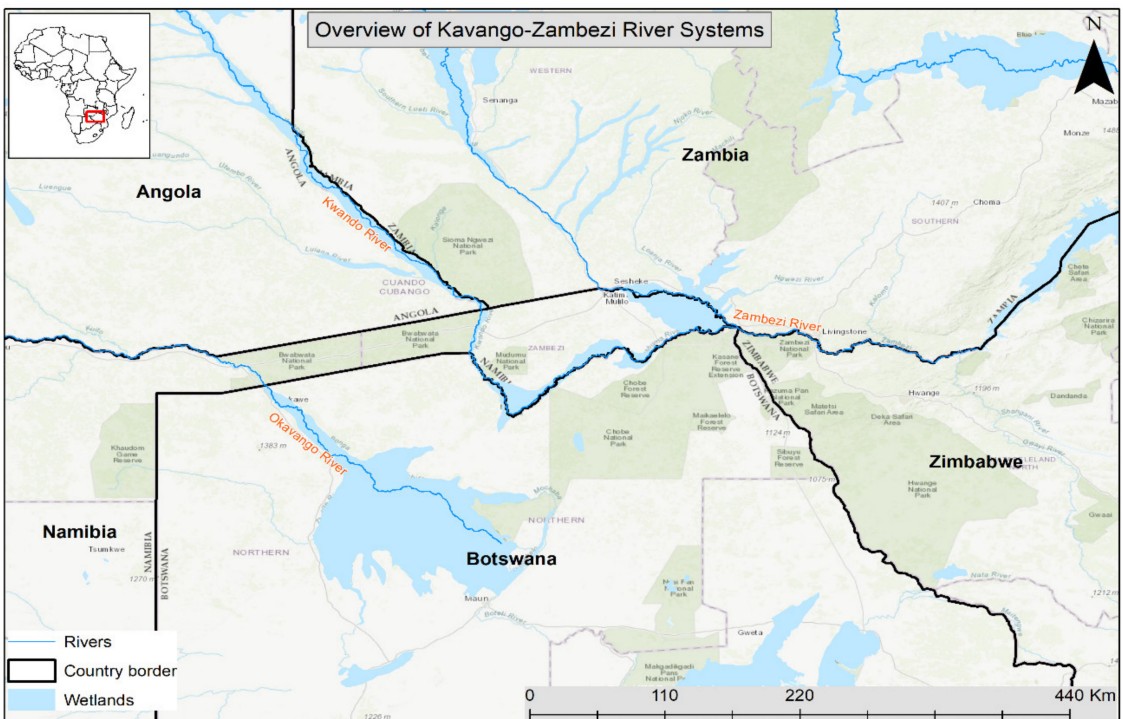

**Figure 1.** Map showing Kavango–Zambezi River system and member countries in the KAZA-TFCA. Source: adopted and modified from Linell [28].

In 2006, several stakeholders, such as Peace Parks, Southern Africa Development Community (SADC), and political actors, such as Nelson Mandela, initiated the idea of a transboundary natural resources management in the world's largest transfrontier conservation area, the KAZA-TFCA [23,28]. It created an enabling environment in which communities of resource users, especially fisheries, could be empowered to participate in the "planning, management, and development of resources through their relevant institutions" [28,29]. This was expected to foster socio-economic development and sustainable

resource use [28]. Although the literature cites successful cases of transboundary natural resources management through collective action responsibilities, the concept may be constrained by incompatible regulations and policies between member countries [29–31]. This assertion calls for a standard methodological strategy in prescribing a conservation plan across the region [32]. This is not an easy task in a social-ecological system characterized by diverse layers of governance [30–32]. It is worthy to note that while management requires implementation at a local level, initiatives made at the national and regional level are critical to link both local and international requirements [33].

The review is based mainly on the cited literature, through examining diverse resource management practices, and our active participation in regional fisheries technical consultation meetings in the region. The focus is on reviewing the governance and management of inland fisheries in the KAZA-TFCA and how it affects the utilization of fisheries resources for food security and biodiversity conservation. The study will provide a window of opportunity to set out management options that can inform conservation practices and policy reforms across the African landscape. It is against this background that this paper seeks to: (i) review the governance and management of fisheries in the KAZA-TFCA in the context of the mentioned socio-political and economic contexts; (ii) examine challenges affecting the implementation of transboundary fisheries management in the region; and (iii) outline possible interventions to optimize the utilization of the fisheries for food security and biodiversity sustainability.

## 2. Methodology

*Study Area*

KAZA-TFCA has one of the largest thriving transboundary fisheries in Africa and holds 85 forest reserves, 11 sanctuaries, 20 national parks, and 103 wildlife management areas [19]. The primary objective was to foster transboundary collaboration in implementing ecosystems and natural resource management in the region that straddles across boundaries [19,23]. Fisheries in KAZA-TFCA are among the most important economic activity for the five partner countries.

The starting point was an electronic search on the FAO country profiles website (http://www.fao.org/countryprofiles/en/) (accessed on 14 June 2021). Two methods of accessing data relevant to the topic of study in the KAZA region included: (i) FAO country profiles; (ii) internet searches using the country name; and (iii) specific keywords related to the management and governance of inland fisheries in the KAZA-TFCA. Each country's profile was downloaded, and the heading on fisheries' constraints and institutional frameworks was used to extract the titles of relevant information. An Internet web search was used to search for information that could not be accessed in the FAO country profile through the references listed within the identified literature. Titles were copied and pasted into a Google Scholar search, the outcome of which provided an array of related literature on the topic of study [34]. The study further used proceedings from the regional stakeholder workshop conducted in 2018 in Namibia to better understand fisheries' management practices at national and regional levels. Key informant interviews were held along the sidelines of the regional stakeholder's workshop. The selection of key informants followed a purposive random sampling technique. The technique is useful for generalizations, since it was not feasible to interview all the participants [34].

A total number of 10 key informants were identified, based on the list of participants obtained from the KAZA secretariat (n = 10). Five (5) of these participants were representative (fisheries officials) from each participating country, three (3) participants were drawn from each participating stakeholder, namely the Namibian Nature Foundation, the World Wide Fund (WWF), and the KAZA secretariat. Two (2) participants were members of participating fisher community conservancies from Namibia and Zambia (Situnga and Simalaha conservancies, respectively). The idea was to verify the accuracy of secondary data sources [35]. The interview questions were based on success, opportunities, and challenges of transboundary fisheries management in the region. This information was

used for analytical work. Finally, our experience and knowledge on the fisheries of the area as researchers and work experience for many years from 2000 up to date as fisheries officers in the area was utilized. The experience helped to broaden the understanding of challenges and success stories of managing transboundary fisheries' resources over time, especially issues related to power relations.

The United Nations's Food and Agriculture Organization Code of Conduct for Responsible Fisheries was employed (FAO, CCRF) as a conceptual framework together with related technical guidelines, such as the precautionary approach. The CCRF recommendation is based on science and experience, it promotes collaboration arrangements among member countries and sharing resources that straddle across boundaries. It's tenets support community participation in the management of resources; it is generally acknowledged as an ideal way of fostering effective governance [36,37]. Furthermore, CCRF provides guidance on policy formulation, which is often a missing link in the operationalization of fisheries activities in most of the developing countries [36].

## 3. The Rationale behind Transboundary Fisheries Management in the KAZA-TFCA

Transboundary fisheries management in the KAZA-TFCA was coined within the context of Southern Africa Development Community (SADC) regional integration and collaboration management of shared resources as a key to the sustainable utilization of natural resources [21]. The term 'transboundary' is defined in the context of international collaboration [31,37] and refers to the development of co-operation across boundaries to enhance the efficiency of achieving objectives of natural resource use and conservation [38,39]. Griffin [37] defined it in a similar way, but he added that transboundary management should benefit the parties involved in the initiative. Social-ecological systems are interconnected in temporal and spatial terms, where the extent of organization and decisions made in a particular place affects the environment and people elsewhere [38]. Consequently, the concept of 'bioregionalism' recognizes that ecosystems do not overlap with political boundaries [37]. As well as the global recognition to promote collaborative arrangements in the management of shared inland fisheries, most threatened inland fishes require distinct habitats for growth, feeding, refuge, and reproduction as such migration institutes an inherent characteristic in their life cycle [39,40]. Therefore, political boundaries must not limit conservation at a larger scale. The idea is to sustain biological and socio-economic gains from aquatic resources that straddle several countries to reduce habitat fragmentation [27,28]. Evidence of increased fishing pressure on inland fisheries shared by various countries has been widely reported [5,17], suggesting that fisheries' conservation efforts can no longer be implemented in isolation. Therefore, the transboundary management approach to aquatic resource is necessary for responsible fisheries.

## 4. Conceptualizing Transboundary Fisheries Management in KAZA-TFCA

The Kavango–Zambezi River system and its network of tributaries is among Southern Africa's most important natural resources [28]. Governance and management approaches for inland fisheries in the KAZA-TFCA is examined to contextualize constraints regarding the implementation of transboundary fisheries management. This information is then used to prescribe potential interventions for transboundary fisheries management in the KAZA-TFCA. According to the FAO Code of Conduct for responsible fisheries, the overall goal of fisheries management is based on sustainable resource utilization [41]. This implies: (i) maintaining the fish stock at the level required to ensure their continuous productivity; (ii) profit maximization of the resource user (fisher); and (iii) optimum employment opportunities for fisher-dependent communities [41]. The simultaneous optimizing of these objectives is rarely achieved [42]. For instance, the maximum exploitation of fisheries for employment opportunities means intensifying fish exploitation, which may abrogate the FAO Code of Conduct on conservation objectives [43]. As a result, identifying a suitable management approach in a shared social and ecological system can be complex due to diverse regulations and conflicting policies among member countries [28]. These regula-

tions are fundamental for the successful management and development of fisheries [26]. While transboundary fisheries management has been widely accepted, it is moderately practiced, based on a classical fisheries management approach, such as fishing ban seasons, gear and mesh size restrictions, among other tools [44,45]. Classical fisheries management in Africa is partly based on ideas drawn from Hardin's Tragedy of the Commons model; the basic assumption towards natural resource management has been to privatize the common resources [46]. Subsequently, Pauly [45] introduced the idea of a quota system as an option to privatization of the commons, essentially to incorporate some form of ethics in management regimes [45].

Apparently, classical management approaches in tropical fisheries are subject to debate on whether anthropogenic activities (artisanal fisheries) have a huge impact on the fish productivity [25,46]. Several researchers [14,25,46] argue that in tropical fisheries, the correlation between human activities and the level of the future fish stock has no considerable impact; it is characterized by uncertainties, where placing a maximum level of fishing effort may not lead to a fixed state of fishing mortality [43]. Thus, fish stock abundance and distribution are largely influenced by environmental parameters, such as river flows that are seasonally driven by the amount of precipitation received in the catchment [43]. Several studies [5,25,45] have shown that classical management perspective is incongruent with the nature and dynamics of inland fisheries in tropical regions, because they do not account for external factors, such as flooding nutrients (humus, cow and wildlife dung, or the build-up of organic matter, which contribute to biological productivity in tropical inland fisheries) [47]. It is further argued that the classic management approach is derived from temperate regions used in marine single-stock management; these classic approaches have based their foundation on a single-species harvest [47]. They are replete with a "hypothetical" equilibrium model for the management of marine fisheries [45]. These models use the idea of maximum sustainable yield (MSY) as the basis of management. The assumption behind MSY is a standard parameter system in fisheries exploitation [48]. Attributes of this concept led to the introduction of mesh size and fishing effort regulations [45]. The classical fisheries management approach restricts the weave size of fishing nets with the goal of restricting smaller fish catches [12]. Over time, evidence has shown that certain fish species in flood plain environs are naturally small, even as adults, for instance, the *cyprinids*. Thus, excluding their catch affects the utilization of the fishery for food security [12,14]. Although the use of classical management approaches has been questioned in tropical inland fisheries, their application in their management is so great that the tenets of the Code of Conduct for responsible fisheries demand that decisions can be made based on experience and existing science to support available management options [41].

## 5. Results

### 5.1. Governance and Management of Fisheries in Zambia

Fisheries and aquaculture development in Zambia is placed under the Department of Fisheries in the Ministry of Fisheries and Livestock. The Fisheries Act No. 22 of 2011 is a principal piece of legislation governing fisheries, together with regulation No. 24 of 2012 [49]. The Act stipulates the establishment of fisheries management areas and the decentralization of fisheries management that includes community involvement in the enforcement of fisheries regulations [50]. These regulations include fish gear, type, mesh size restrictions, and the issuance of fish licenses to regulate fishing effort and access to the fishery areas [49]. However, the regulations do not specify the number of fishing nets a fisher can use [22,33]. The fish closed season is annually imposed from 1st December to the end of February as a way of protecting the fish stock during the production phase to ensure successful recruitment [49]. This regulation has been largely criticized in tropical fisheries due to insufficient data in support of it [51,52], which may also be relevant for inland fresh water fisheries. Several researchers [24,25,51] have argued that floodplain fish species have different life cycles and natural fluctuations in recruitment are triggered by flood regimes [25].

The fisheries sector in Zambia presently does not have a stand-alone policy on the management aspect of fisheries and aquaculture [42]. The policy statement is covered by the National Agricultural Policy (NAP) (2004–2015) that governs the development of the agriculture sector in Zambia [42]. A close analysis of interventions in the NAP shows more emphasis on crop production than on fisheries development. Furthermore, the policy focuses more on maximizing fish production and employment opportunity than on sustainable utilization of the fisheries resource [43]. This approach poses a management dilemma due to a lack of adequate fisheries data on which to base effort restriction. While fisheries co-management is enshrined in the Zambian Fisheries Act No. 22 of 2011 [49], the Act does not explicitly exhaust matters related to transboundary fisheries management and lacks a comprehensive legal framework to support legal certainty of the fisheries development agenda [33]. The lack of an adequate national fisheries policy in Zambia seems to explain the reasons why the fisheries sector has received very little attention in terms of funding [16]. This shortfall has, in turn, resulted in a failure to bring about the institutional reforms necessary to enhance the utilization of the fisheries' resources both at local and international levels [14]. These claims are supported by the WorldFish Center [16] report, which revealed a low priority of data collection accorded to the Zambia fisheries.

The system of fisheries governance on the Zambezi River in Zambia is characterized by two parallel governance structures, the central government and the traditional authorities, which are often at odds with each other [53]. Tension and disagreement have often arisen between the central government and the traditional authorities over leadership roles in the governance of the fishery [26]. The main attributes of the local leadership in the area are: (i) fishing access restrictions; (ii) enforcement and punitive measures; and (iii) traditional fishing wardens [26]. The system of governance by the traditional authorities exercises authority that allows access rights for the kingship where migrant fishers get authorization from local headmen to settle in seasonal camps during the fish ban period [26]. According to a key informant from Simalaha conservancy, these conditions are likely to stymie the development of a transboundary fisheries management regime, obviously because of the existing parallel and conflicting governance structures in the fishery [26]. The contentious issues are on the aspect of a fish ban season; the traditional leadership does not fully support the idea of a fish ban due to limited livelihood options and failure by the government to recognize access rights [26].

### 5.2. The Zimbabwean Context

The governance of fisheries in Zimbabwe is under the Zimbabwean Parks and Wild-life Management Authority (ZPWMA). This institution falls under the Ministry of Environment and Natural Resources [14]. The Authority bears the responsibilities for wildlife (terrestrial) and fisheries management in terms of the Wildlife Act–Chapter 20:14 of 1996. The Act outlines fisheries regulations, aquaculture, and the development and control of the fishing industry. It also stipulates financial provisions, enforcement, penalties, and offences, including general provisions. Access to fishery areas in Zimbabwe is regulated through a license system [11]. The annual system of licensing stipulates where one can fish and how many gillnets a fisher can use [33]. Fish gear and mesh size restrictions are regulated to control fishing effort. The annual licenses are issued by the ZPWMA upon payment to the ZPWMA. This is done in consultation with local authorities (District Councils) [11]. This approach has often created user conflicts between government authorities and local resource users whose livelihoods depend primarily on access to the fishery resources [33]. The resource users appear to play no significant role in the management of the fisheries' resources [54], suggesting that the fisheries governance and management in Zimbabwe is a centralized 'top-down' approach. The scope of involvement by local resource users in the process of decision making with regard to management and utilization of the fisheries' resources is very low, despite the fact that the participation of resource users in management of their resources is acknowledged as a requirement for sustainable development of the fisheries' resources [54]. In 1995, the FAO recommended that active engagement of resource

users and stakeholders' participation in fisheries management is among the major principles in the implementation of Code of Conduct for Responsible Fisheries [41]. Sustainability of inland fisheries' resources in Zimbabwe has received wide criticism [43,54], based on the argument that the top-down management approach has resulted in regulations that are inappropriate for local resource users [54].

Like many other countries in Southern Africa, Zimbabwe does not have a national fisheries policy [14]. This, according to Mhlanga and Mhlanga [54], explains why the country lacks appropriate interventions related to the role of the fishery sector in the national economy [14]. This suggests that the lack of a national fisheries policy in Zimbabwe has hindered institutional reforms that would have activated stakeholders' participation in decision making regarding fisheries management at a local level [14]. Viswanath et al. [55] observed that the lack of a clear fisheries policy direction in Zimbabwe has resulted in a weak management system, which is insensitive to local conditions and less likely to meet its own objectives (that is, a sustainable utilization of the resource). The principal piece of legislation governing fisheries activities (the Parks and Wildlife Act) appears to be more skewed towards management of wildlife than fisheries. As a result, fisheries are not accorded the necessary reforms to unleash their potential [14]. While Zimbabwe is a signatory to regional and international protocols on responsible fisheries, the commitment to these protocols is low and the process of institutional reforms in fisheries has stagnated owing to lack of policy direction.

*5.3. The Botswana Context*

The management of fisheries' resources in Botswana is under the purview of the Environment, Wildlife and Tourism Ministry [14]. The sector is run under a division within the Wildlife sector. The Fisheries Protection Act 42 of 1975 is the principal legislation governing the fisheries activities in Botswana, together with other pieces of legislation, such as the Fisheries Protection Regulations of 2016 and the Statutory Instrument of 2015 [14]. The latter regulates fishing effort with gear and mesh size restrictions [14]. Permissible fishing gears in Botswana include long lines, gillnets, and hook and line. The fish ban season is implemented for two months between January and February to protect fish breeding and recruitment during the period. However, the timing of the fish ban season remains a challenge, as it does not coincide with the fish ban season in Namibia and Zambia, where the ban is observed for three months between December to February [53]. Botswana does not have a fisheries policy on which management interventions are based [14], which suggests that management measures made without a policy may not adequately address the management concerns of the fisheries sector due to a lack of policy direction. The fisheries sector in Botswana falls under a Wildlife Management Authority whose management philosophy is more focused on conservation than sustainable exploitation of the fisheries for food security as prescribed by FAO Code of Conduct for responsible fisheries [38]. This system of governance appears to focus on the development of wildlife to sustain the tourism sector rather than the conservation of the fisheries resources for food security [14]. The problem is amplified by the lack of clear policy direction to bring about appropriate technologies and informed decision making. The major challenge is that investment in the research of the fisheries to generate the data required for development is not adequate [14], as well as not having a stand-alone policy to provide oversight and direction on fisheries management and development. The current management approach is insensitive to the access rights of fishing communities [54–56].

*5.4. The Namibian Context*

Governance of inland fisheries in Namibia is under the Ministry of Fisheries and Marine Resources (MFMR) [57]. The Inland Fisheries Resources Act No. 1 of 2003 is the principal piece of legislation governing the inland fisheries in Namibia. [58]. The Act is developed in the contest of the white paper on the Responsible Management of Inland Fisheries in Namibia [58]. Namibia is one of the few countries with a policy dedicated

exclusively to inland fisheries. The philosophy followed is that different management approaches are devised for different rivers systems due to the diverse nature of some of these systems. Furthermore, the interest of the subsistence households on the availability of fish as food security is given priority and the need to control any commercialization of the resource. The Act provides for the conservation of aquatic ecosystems and the sustainable development of inland fisheries' resources [58]. The Namibian Fisheries Act recognizes the transboundary management of fisheries of a shared river system [58]. Furthermore, the management system provides for fisheries reserves, defined, regulated, and enforced by local resource users. A closed fishing season for the rivers systems, exclusively for the Zambezi Region is in place from the 1st of December to the end of February, similar to Zambia. The current management approach to fisheries on the Namibian side of the Zambezi River is administered by both the traditional authority and the central government [25]. The management strategies include effort restriction in the form of fishing gear restriction and type allowed, the number of gillnets allowed per fisherman, mesh size restriction, and the fishing method practiced [58,59]. Fisheries reserves are not necessarily no-take zones and may allow fishing with specific rules according to the demands of the local communities. These rules, however, must be within the framework of the Inland Fisheries Resources Act [60]. A Standing Operating Procedure (SOP) was developed, endorsed by the Ministry of Fisheries and Marine Resources, to give guidance when creating fisheries reserves [60–63]. Furthermore, the tackle box for community fisheries reserves highlight the step-by-step approach to establish community-based fisheries reserves [63]. The traditional leadership plays a vital role in the management of the fisheries resources [64] and needs to be considered for future management approaches. Regulations place emphasis on fish gear restrictions and access rights to particular fishing grounds; they do not restrict access to any fishing area during the high-water period [65–67]. However, permission is needed for fishing in secluded areas during periods of low water levels [68]. The Namibia Nature Foundation conducted numerous frame surveys in the Kavango, Kwando, Zambezi, and Chobe Rivers to gain insight into the value fish play in these communities, how fish are managed through local communities, and into the livelihood strategies communities pursue. Although the Namibian inland fisheries policy of 1995 has been reported to have a more influential profile due to its transparence attributes [12], the system of community participation in fisheries management appears to be moderated.

*5.5. The Angolan Context*

Inland fisheries on the Kavango–Zambezi rivers system in Angola have remained underdeveloped partly due to lack of a policy to guide governance processes. The 27-year post-civil war that destroyed infrastructure had dragged institutional processes to establish policies and regulations. [69]. The scanty information available has revealed institutional problems and processes in explaining the underdevelopment of the inland fisheries sector in Angola. Key informant interviews from both the KAZA secretariat and a fisheries official from Angola revealed that the management of inland fisheries in Angola is managed under the jurisdiction of aquatic biological resources of 2004 (LRBA), Law-A/04 of 8 October 2004 [62]. The law focuses on the protection and sustainable use of aquatic resources. The role riverine communities play in the protection of these aquatic resources is recognized. Recently, policies moved towards a more community-based attempt where communities have the right to manage their natural resources [69]. Managers still lack knowledge of community-based approaches in fisheries management. This lack of knowledge is still seen as an obstacle in establishing fishing zones that should be managed by communities for their own benefit. The Presidential declaration 139/13 of 24 September 2013 regulates the Law on Aquatic Biological Resources with regard to inland fisheries [70]. The purpose is to establish rules governing the conduct of inland fishing activities in the inland waters of the Republic of Angola.

Some regulatory mechanisms in the law are the development of management plans for all inland fisheries, the defining of fishing zones and protected areas, and the setting of



fishing effort limitations, minimum sizes of fish species, prescribed mesh sizes, and fishing methods. The law further makes provision for the consultation with fishermen associations and community organizations and the devolution of power to the local level [70]. Provision is also made for closed seasons, although no current closed fishing season is in place [71]. Recent training of staff on sampling and survey protocols were done between Namibia and Angola with the aim to establish a shared database in the future for management purposes. Similar activities were performed in the establishment of fisheries reserves with the aim to have these reserves across borders, managed by communities on both sides of the river [65]. Most investment in subsistence fishery is in the coastal fisheries with very little channeled to inland fisheries. While Angola appears to have sustainable fisheries strategies and plans, these have not been implemented due to lack of a policy framework to stimulate policy debate on inland fisheries and guide governance processes [71].

## 6. Discussion

Classical management approaches are widely used to manage fisheries in the region, which, as shown in this paper, can be problematic in various aspects. It is argued that countries in the KAZA-TFCA have different management approaches and regulations characterized by inadequate policy [56], and that assessing several regulations, policies, and local institutions in the region is cardinal before scaling-up conservation measures [71]. This paper further argues that the implementation of a transboundary fisheries management regime throughout the KAZA-TFCA has largely failed. We hold the idea that transboundary fisheries management has shown positive results as a management concept in diverse sectors of natural resources governance, including fisheries [31,36], and theoretically, it shares a number of advantages of bottom-up management [30]. The fundamental difference in the regulatory frameworks, management approaches, and inadequate policies governing the fisheries' resources in the region has created a stalemate for the successful management of aquatic resources. This has serious policy implications for transboundary fisheries management in the region and elsewhere in the world [72]. Implementation of transboundary management as a blueprint without harmonizing the fisheries regulations, policies, and management approaches is a constraint towards conservation and a sustainable livelihood for local residents. This may further limit conservation at a larger scale because of incompatible fisheries regulations that may result into wide-spread free-riding activities [46,56].

### 6.1. The Dilemma of Transboundary Fisheries Management in the KAZA-TFCA

The fisheries regulations that exist across the countries in the KAZA-TFCA make the actualization of transboundary initiatives difficult. For instance, within the KAZA-TFCA, recreational fisheries in Botswana, Namibia, and Zimbabwe are allowed throughout the year, even during fish ban season, while in Zambia, recreational fishing is prohibited during the fish ban period. This conflicting regulation over the utilization and accessibility of the fishery resources during the fish ban has affected co-operation of shared management responsibilities among member countries, especially at local level [73]. Similarly, the implementation of a fish ban season of three months observed in Zambia and Namibia from December to February the following year, essentially to cover for the fish breeding season for most fish species in the Zambezi River system [24], is in conflict with the two-month ban enforced in Botswana between January and February, while Zimbabwe does not observe this ban at all [14]. The situation creates considerable challenges for transboundary fisheries management on the Kavango–Zambezi River system owing to free-riding activities by fishers in countries that do not implement a fish ban [35]. Furthermore, the efficiency of the fish ban season to allow fish breeding and recruitment is questionable because the timing does not coincide with the breeding period of most of the targeted fish species, such as the Tilapine fish species. Most of these breed around October–December. This situation is perceived to cause resentment among various stakeholders and to reduce the overall effectiveness of transboundary fisheries management at the local level [24].

Furthermore, Zambian fisheries policy emphasizes the maximum fisheries production to meet the food requirements of the growing population and to secure employment opportunities for rural communities [43]. As a consequence, many small-scale fishers who can afford entry into the fishery are allowed unfettered access, without gillnet limitations [43]. This approach may lead to over-exploitation, "Tragedy of the commons", and compromises the sustainable utilization of the fisheries' resources as prescribed by the FAO Code of Conduct for responsible fisheries [41]. This further conflicts with the fishing regulations of neighboring countries, such as Zimbabwe, Namibia, and Botswana, where fishing effort is regulated by limiting the number and size of nets a fisher is allowed to use [46]. Conversely, the Zimbabwean fisheries regulations appear to restrict and regulate the number of gillnets a fisher may use and to stipulate where one can fish [43,54]. Artisanal fishers are restricted entry to certain fishing grounds [43]. This kind of management approach appears to focus more attention on recreational fisheries and wildlife activities to support eco-tourism than on utilization of the fishery for food security. This is despite the fact that tourism conservation is likely not to create better opportunities and equal access to resources for rural riverine communities whose livelihoods depend on fishing [33].

The differences in management approaches have generally made it difficult to actualize transboundary initiatives, as different countries within the region appear to pursue diverse social economic models of development. These differences affect coordination and collaboration among participation countries at a local level [73]. For instance, the five countries in the KAZA-TFCA have pronounced policy statements in favor of transboundary fisheries management, but the existing divergent economic policies among member countries hinder meaningful implementation of equitable shared responsibilities [33,43]. Key informant interviews from both Zambia and Zimbabwean officials revealed that there are differences with regard to mesh size prescribed by Zambia fishing regulations and of Zimbabwe. These findings are further highlighted in the 2011 Zambia/Zimbabwe fisheries Frame Survey report, which showed that gillnets of mesh sizes below 63 mm (2.5 inch) were also reported to be used in Zambia. The nets increased from 51 in the 2006 Survey to 189 in the 2011 Frame Survey [43]. The use of these nets (below 63 mm) is prohibited in Zimbabwe. These differences in the regulations of inland fisheries complicate transboundary fisheries management at the local level and creates a social dilemma for a person's rational behavior, in which the absence of limits to maximize short-term personal gain may result in environmental degradation [74]. It is difficult to realize the fisher's incentive to cooperate over resource conservation as opposed to harvesting as much as possible since fish, by nature, are fugitive [44]. The fish you do not harvest today are most likely to be harvested by someone else tomorrow [74,75], leading to the tragedy of the commons.

Given the above scenario, it is apparent that a lack of compatibility of fisheries regulations between nations in the KAZA-TFCA is a huge challenge for the successful implementation of a transboundary management regime. The problem is exacerbated by inadequacies and the lack of national fisheries policies among the member countries [14,76]. This is particularly evident in the failure to optimize usage of the many dams in Zimbabwe and south-eastern Botswana for fisheries production [14]. It is acknowledged that the overriding objective of fisheries management is the biological sustenance and utilization of fisheries' resources to support the economic and social wellbeing of fishers [67]. Consequently, the development of transboundary fisheries management must be guided by a policy framework to guide the governance and utilization of the resource for food security at both national and international levels [68]. Apparently, the management of fisheries in Botswana and Zimbabwe is run under a division within the Wildlife Management Authority. The challenge with this institutional arrangement is that both wildlife terrestrial and fisheries resources are managed for conservation to sustain tourism, since recreational fisheries integrates so well with tourism [14]. This management approach denies resource users real opportunities to utilize the fisheries for food security as prescribed by FAO Code of Conduct for responsible fisheries [14,41]. The placement of the fisheries division under the wildlife sector has a negative bearing on the performance of the fisheries sector because

of the frequent and unpredictable institutional changes in the wildlife sector, which is often influenced at the global level [14]. The conditions whereby fisheries management is a responsibility of the Parks and Wildlife Authorities is a compromise as these sectors are not 'development' orientated [14,68]. In other southern African countries where fisheries have a relatively established institutional profile (such as Malawi and Mozambique), transboundary fisheries management has been relatively successful [31], for instance, the establishment of a vibrant transboundary fisheries management scheme on Lake Chiuta, shared between Malawi and Mozambique. The achievement of the recently coordinated regulations in that fishery was a prerequisite to a successful implementation of a transboundary fisheries management in the lake [31]. While the idea of transboundary fisheries management has been widely recognized since the 1980s, it has recently grown in prominence through institutional reforms and collaborative initiatives [77]. For instance, a more creative partnership between the United States of America (US) and Mexico illustrates the creation of joint solutions to environmental challenges of a shared Colorado River basin [77]. In the Laurentian Lakes, where the United States of America and Canada have set up management structures to address policy and environmental issues, a harmonization of policies played a key role in establishing management structures [78]. While in the African Great Lakes region, a collaboration initiative is newer. Diverse governance systems among the riparian states sharing Lake Victoria, for example, have been among the major hindrances to the effective management of the lake ecosystems [78].

The fundamental variations in the utilization of fisheries' resources in the KAZA-TFCA show dissimilar behavior in resource exploitation and conservation, thereby making conservation prescription very difficult [35,68]. Overall, these accounts suggest how coordination and harmonization of regulations between partner states could be an essential aspect in making transboundary fisheries management sustainable on a long-term basis. According to Ostrom [75], the governance of social and ecological systems will have a positive outcome only if sufficient conditions are incorporated in the management aspects. As such, Ostrom identifies eight interrelated design principles for the effective management of resources held in common. Principle design numbers six, seven, and eight are based on the existence of a system of conflict resolution mechanism, the recognition of rights to self-organize, and a framework for co-ordination among relevant stakeholders, respectively [74,75]. These principles have the scope of application for transboundary fisheries management in achieving collective responsibilities in the KAZA-TFCA and elsewhere [40,79]. The absence of the aforementioned principles may lead to user conflicts, non-compliance to rules governing the resources, and free-riding activities [69]. The interactive effects of abrogating these principles in the management of transboundary fisheries contradicts the sustainable utilization of fisheries' resources as prescribed by FAO Code of Conduct for responsible fisheries and may weaken the initiatives of transboundary fisheries management both at local and international levels [33,80].

*6.2. Prospects for Transboundary Fisheries Management in KAZA-TFCA*

Several studies, so far, show that fisheries in the KAZA-TFCA have the potential to contribute to the food security and nutritional status of over one million people who regularly eat fish in the region [21,24,56]. A standard approach to support transboundary fisheries management is necessary to guarantee future prospects. FAO and SADC [81,82] call for transboundary fisheries management activities through coordinated and cooperative arrangements in the Zambezi River Basin. It is generally acknowledged that transboundary initiatives can enhance security, peace, and the long-term stability of people and resources. These goals can be actualized by recognizing regional and international protocols to which all member countries discussed here are signatories. Management of transboundary inland fisheries in Southern Africa, to a larger extent, stems from the SADC protocol of 2001 whose principals were founded on the United Nations Convention on the Law of the Sea 1982 (UNCLOS) and the principles of the FAO Code of Conduct for Responsible Fisheries (CCRF) (FAO, 1995) [41,83,84].

In light of these findings, we propose to use the FAO Code of Conduct for Responsible Fisheries (CCRF) of 1995 as a conceptual framework to secure the sustainable utilization of fisheries' resources in the KAZA-TFCA as a first step. The CCRF draws from experience and science-based evidence in designing international instruments, policies, and plans to support responsible fisheries management. It further declares that, "States should co-operate at global, regional and sub-regional [level] to promote management and conservation to ensure responsible fishing" [41,60]. The CCRF provides important guidelines for developing good management practices and policies for sustainable fisheries and aquaculture development [83]. Furthermore, the CCRF provides for community participation in the governance and management of natural resources, an aspect that offers a window of opportunity to incorporate indigenous knowledge in resource management [85]. Indigenous knowledge is context specific, and local resource users often desire to use concrete knowledge in time and space [41,68,86]. Although the CCRF is a voluntary code, its guiding principles are internationally acknowledged in the management of fisheries [70]. The CCRF is complemented by several other technical guidelines on how to implement specific provisions, for example, the precautionary approach to capture fisheries [84]. Nevertheless, designing regional and international agreements into practical strategies will require commitment and backing in many cases from interested parties, in this case, member states within the KAZA-TFCA who have already signed a memorandum of understanding for regional collaboration in the management of transboundary natural resources [23]. Regional institutions, such as SADC, are critical vehicles to influence regional stakeholders. The SADC Protocol on Shared Water Bodies and the SADC Treaty on Management of Watercourse Systems provide an opportunity for such stakeholder coordination consistent with the mission statement of the KAZA-TFCA [59].

The transboundary fisheries management challenges reviewed in this paper would be addressed by fulfilling certain key attributes, highlighted above (Table 1). For instance, (i) a lack of cooperation among member countries and stakeholders in KAZA-TFCA would be addressed by coordination and collaboration amongst the countries sharing the fisheries resources [87,88] Collaboration among member countries will create a platform for sharing knowledge, technology, expertise, and financial resources towards meeting the regional development agenda [87,89]; (ii) the inadequate institutions and inconsistence in management approaches would be addressed by fulfilling the attribute of having strong institutional frameworks. FAO and CCRF provide technical guidelines on policy development and essential institutions for fisheries management [78]. Policies are intended to guide governance processes and highlight the role and responsibilities of all stakeholders who may be key towards having strong transboundary collaborations [89].

**Table 1.** The key attributes essential for successful implementation of transboundary fisheries management.

| No. | CCRF Attributes | Rationale |
| --- | --- | --- |
| i | Co-ordination and collaboration | CCR provides policy harmonization and the integration of management approaches. In this case, countries in KAZA-TFCA can adopt to facilitate informal mechanisms for collaboration and conflict resolution in response to the objectives of transboundary cooperation. |
| ii | Strong institutional framework | CCRF stresses the need to develop institutional frameworks and policies which provide guidelines on the utilization of the fisheries' resources for food security, whilst taking into account compatible measures within and beyond the national jurisdiction to ensure responsible fisheries. |
| iii | Social-ecological approach | CCRF Promote biodiversity conservation and availability of fisheries' resources in sufficient quantities for present and future generations in the context of food security and poverty alleviation. |
| iv | Stakeholder's participation | CCRF emphasizes the need for community participation in the governance of fisheries' resources. The assumption is that if communities are involved in conservation, the benefits they receive will create incentives for them to become good stewards. |

**Table 1.** *Cont.*

| No. | CCRF Attributes | Rationale |
|---|---|---|
| v | Precautionary approach | CCRF provides guidelines to prevent and mitigate potential threats to marine and inland fisheries. |

Adopted and modified from FAO [41,83].

To avoid conflicts among the stakeholders, attention has to be premised on the fact that the decision-making process incorporates all the stakeholders involved and that the strategy is multidisciplinary, i.e., involve fish traders, fishers, officers, value chain actors, and several other resource users to sufficiently integrate indigenous knowledge and create a sense of ownership and adherence [44]. Furthermore, Campbell and Olson [90–93] argued that major decisions and priorities regarding which resources to enhance for which institutions are not arbitrary, they rather mirror the interest of the most powerful groups whose power is interceded through political, economic, and social institutions. In light of this, political influence will be essential in facilitating an enabling environment for policy reforms in transitioning to CCRF [43]. Similarly, other challenges discussed in this paper would be addressed by the attributes highlighted in Table 1. Furthermore, CCRF provides for a precautionary approach towards the management of fisheries' resources, this is particularly important for KAZA-TFCA where data on which to extrapolate decisions are inadequate. Precautionary approach is widely considered as a guiding tool for policy and management decisions in times of uncertainty, as it shifts the decision-making process to anticipate, prevent, and mitigate threats [38,78]. Several studies, e.g., Coll et al. [80,81], have exalted the efficiency of CCRF in fisheries' resource management, by comparing compliance with the CCRF to changes in five ecological indicators, which quantifies the ecosystem effects of fishing. The loss in production index and the related probability of sustainable fishing index were tested in different regions. The results indicated that countries with higher levels of compliance with the CCRF experienced a decrease in production index loss and an upsurge in fisheries sustainability from the 1990s to 2000s. The study concluded that the implementation of the CCRF resulted into positive ecological outcomes. However, CCRF must not be seen as a universal panacea to deal with all the problems related to transboundary fisheries management; more studies and research are still needed to learn about better conditions leading to the implementation of a successful transboundary fisheries management.

**7. Conclusions**

This paper has presented a review of the fundamental differences with regard to a governance and management perspective of transboundary fisheries in the KAZA-TFCA. The study has illustrated why uniform conservation prescription matters in resource conservation. The focus of the study has been to argue that member states in the KAZA-TFCA have substantially different management approaches. With the exception of Namibia, none of the KAZA participating countries has a fully developed fisheries sector policy to guide governance roles and processes. Regulatory frameworks are constrained by inadequate institutions, which requires capacity building to strengthen coordination. This gap appears to undermine the sustainable utilization of the fisheries' resources in the region for food security, and subsequently, a transboundary management regime. They further create differences in the way the fisheries resources are accessed, managed, and utilized. The debates about fisheries governance and management are essentially to enhance livelihood opportunities for riparian communities. The study provides useful insights into the factors constraining transboundary fisheries' resources to transform the future. The findings have policy implications over the need to have a fisheries policy in transitioning towards sustainable fisheries management and enhance food security and environmental sustainability. The immediate target should aim at stabilizing the fishery by having a coordinated response involving all the partner countries and to harmonize fisheries' regulations and management approaches. If these fisheries' resources are managed sustainably, inland fisheries can contribute towards the attainment of the United Nations' Sustainable Development Goals (SDGs) of biodiversity conservation and poverty

alleviation by 2030. Based on FAO reports, SDGs can be realized through poverty eradication by 2030. The first and second SDGs emphasize the need to "eradicate poverty" and "hunger" [81], which can be accomplished through improved rural development initiatives in fisheries and fish farming development. This is crucial and very urgent in southern Africa, where a United Nations (2016) report indicated that poverty is widespread, with about 40% of the population surviving on less than USD 1.95 per day in 2012.

**Author Contributions:** I.I. wrote and conceptualized the paper. S.A. reviewed, edited, and supported the conceptualization of the paper. W.S. provided expert guidance on the methodology and structure of the paper. C.H. reviewed and wrote the paper. All authors have read and agreed to the published version of the manuscript.

**Funding:** This review paper is made possible in part by the Rhodes University and Zambia Aquaculture Enterprise Development Project (ZAEDP). Partially funded by the European Commission E€OFISH Programme for the project "Strengthening Community Fisheries in KAZA" implemented by the Namibia Nature Foundation, Windhoek. The contents of this review paper are the sole responsibility of the authors.

**Institutional Review Board Statement:** The study was conducted under the guidance and approval of the Rhodes University Human Ethics Committee (RU-HEC). Reference number 2020-1571-4696.

**Informed Consent Statement:** Informed consent was attained from all parties involved in the study.

**Data Availability Statement:** The data presented in this study are available online; certain information can be provided on request from the corresponding author, owing to ethical restrictions.

**Acknowledgments:** The authors wish to acknowledge K. Mosepela from the Botswana University of Agriculture and Natural Resources, for providing specific data and expert advice on inland fisheries in the region. Furthermore, our acknowledgements go to Harris Phiri and Evans Mutanuka, from the Department of Fisheries in Zambia, for providing gray literature and insights on the management of inland fisheries in Zambia, particularly on the Zambezi River system.

**Conflicts of Interest:** The authors declare no conflict of interest.

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
