# Peer review of "Transboundary Fisheries Management in Kavango–Zambezi Transfrontier Conservation Area (KAZA-TFCA): Prospects and Dilemmas"

_sustainability, doi:10.3390/su15054406_

Round 1

Reviewer 1 Report

The paper entitled Transboundary Fisheries Management in Kavango-Zambezi Transfrontier-Conservation Area (KAZA-TFCA): Prospects and Dilemmas aimed to figure a possible solution in fisheries management in a region in which five countries politics and governances completely different act according to their possibilities. The manuscript fails in presenting a real possible solution to the conflict, instead the authors present a short revision of governance in Zambia, Zimbabwean, Botswana, Namibia and Angola. The heterogeneous socio-political context among these five countries do not let them to stablish a real solution. The author identified a two introduces species with aquaculture purposes that increase the problem they aboard. The fish Oreachromis niloticus and the crustacean Cherax quadricarinatus. The authors concluded that there is no real institution to prompting the fisheries governance in the region. The main goal of the paper was not addressed because there is a no real hypothesis to be solve.

Author Response

Reviewer report one has been uploaded

Reviewer 2 Report

The paper is good, well written and scientifically based. However, the work needs to be contextualized to an international context, the work is very specific to a place in its discussion, I recommend covering the discussion more with an example in the world.

Author Response

The report for reviewer number two has been uploaded

Reviewer 3 Report

The author has conducted a study entitled "Transboundary Fisheries Management in Kavango-Zambezi 1 Transfrontier- Conservation Area (KAZA-TFCA): Prospects and 2 Dilemmas".

I have completed my evaluation of your manuscript. My comments on the manuscript are as follows:

This work is very profitable for the government in an integrated fisheries management effort. However, the author should enrich this paper with data from the author's experience and knowledge about fisheries in the area so that the statement is valid

In the results and discussion section, the author should include the results of the research (results of interviews) that have been analyzed and discussed. Avoid subjective statements without data.

Methodology:

1. The research objectives are stated at the end of the introduction

2. The research method must be sharpened and preferably explained in the form of a flow chart so that the stages become clear and detailed

3. The interview link should be made in the Google form

4. How many institutions and participants participated in the regional workshop? Does the key informant represent the study area? Is the number of key informants in the research sample following the sampling rules?

5. How to get accurate and valid data? Indicate by appropriate analysis

6. How to analyze and validate the literature review from web searches?

7. How is the analysis used from the findings collected (results of interviews)?

Sections 3 and 4 should be combined in the introduction by emphasizing the crucial points of the research.

Results and Discussion

1. Careful analysis of the interventions in the NAP has shown more emphasis on crop production than fisheries development. What data can prove this?

2. Fisheries Governance and Management in Zambia (Zimbabwe, Botswana, Namibia, Angola) should be made in an informative table related to governance, including its shortcomings (-) and strengths (+), and made a weighting scoring analysis in determining the best governance of the four Fisheries Management.

I recommend reconsideration the manuscript following major revision. When revising the manuscript, please carefully consider all issues mentioned in the comments.

Author Response

Reviewer report number Three (3) has been uploaded

Reviewer 4 Report

Transboundary Fisheries Management in Kavango-Zambezi  Transfrontier- Conservation Area (KAZA-TFCA): Prospects and Dilemmas by Imbwae et al provides fishery management in Africa.

This study is important for the management of transbaoundaries between the african contries. The challanges and difficulties were clearly presented and it will be beneficial for all the countries involved.

There are some spelling mistakes or brakes in some words in the text those can be improved. It has 85 forest reserves, 11 sanctuaries, 20 national parks, and 103 wildlife management 97 areas and the management of such big ecosystems could be very difficult and this review presents the situation and provides clear solutions.

The manuscript especially in the discussion section is very long and it should present clear solutions to everyone. I suggest authors to make such shorteenning in the discussion and conclusion. Table 1 can be remodified accordingly.

Author Response

Reviewer report number four (4) has been uploaded

Reviewer 5 Report

This paper is well researched and well organized. The topic is new and I believe this paper will in some way fill up the gap between the current literature and practice. There are some small mistakes in the paper that should be fixed before this paper is acceptable for publication.

1.      Abstract, there are quite a number of miss words such as “com-munities”; “to-wards”; “govern-ance”; “re-gional”. These should be fixed.

2.      P.2, What is “SADC” stand for? You need to make sure providing full name for this abbreviation when you adopt it first time in the paper.

3.      P.3, line 101. The sentence needs full stop.

4.      P.6, lines 251-252, “…both at local and international level” it should be “both at local and international levels”.

5.      P.11, Lines 496-497, Lines 533, “…both at local and international level” it should be “both at local and international levels”.

6. P13, Line 582,“resource(Table 1Needs full stop. Line 589, “processes. they provide a foundation on which …” it should be “processes. They provide a foundation on which …”

Author Response

Reviewer report number five has been uploaded

Round 2

Reviewer 1 Report

The authors made improvements that merit the acceptance in its actual form.